



**The role of coccoliths in protecting *Emiliania huxleyi* against stressful light and**

**UV radiation**

**Running Title:** Photoprotective role of coccoliths in *Emiliania huxleyi*

Juntian Xu[1,2], Lennart T Bach[3], Kai G Schulz[3], Wenyan Zhao[1], Kunshan Gao[1*], Ulf

Riebesell[3]

[1]State Key Laboratory of Marine Environmental Science, Xiamen University, Xiamen,

Fujian, 361102 China;

[2] Key Laboratory of Marine Biotechnology of Jiangsu Province, Huaihai Institute of

Technology, Lianyungang, Jiangsu, 222005 China;

[3]GEOMAR Helmholtz Centre for Ocean Research Kiel, Düsternbrooker Weg 20, Kiel,

24105 Germany

*Author for Correspondence: ksgao@xmu.edu.cn (Kunshan Gao)



## Abstract

Coccolithophores are a group of phytoplankton species which cover themselves with small scales (coccoliths) made of calcium carbonate ($CaCO_3$). The reason why coccolithophores form these calcite platelets has been a matter of debate since decades but has remained elusive so far. One hypothesis is that they serve a role in light/UV protection, especially in surface dwelling species like *Emiliania huxleyi* which can tolerate exceptionally high levels of solar radiation. In this study, we tested this hypothesis by culturing a calcifying and a non-calcifying strain under different light conditions with and without UV radiation. The coccoliths of *E. huxleyi* reduced the transmission of visible radiation (400-700 nm) by 7.5%, UV-A (315-400 nm) by 14.1% and UVB (280-315 nm) by 18.4%. Growth rates of the calcifying strain (PML B92/11) were about 2 times higher than those of the non-calcifying strain (CCMP 2090) under indoor constant light levels in the absence of UV radiation. When exposed to outdoor conditions (fluctuating sunlight with UV radiation), growth rates of calcified cells were almost 3.5 times higher compared to naked cells. Furthermore, relative electron transport rate was 114% higher and non-photochemical quenching (NPQ) 281% higher in the calcifying compared to the non-calcifying strain, implying higher energy transfer associated with higher NPQ in the presence of calcification. When exposed to natural solar radiation including UV radiation, maximal quantum yield of photosystem II was only slightly reduced in the calcifying but strongly reduced in the non-calcifying strain. Our results reveal an important role of coccoliths in mitigating light and UV stress in *E. huxleyi*.




**Key words:** coccoliths, *Emiliania huxleyi*, light protection, growth, photosynthetic
performance, UV radiation

## 1 Introduction

Coccolithophores are a group of marine phytoplankton species which are able to
precipitate $CaCO_3$ in the form of small calcitic scales (coccoliths) surrounding the
organic part of the cell. They contribute about by 1-10% to marine primary production
(Poulton et al., 2007) and approximately 50% to pelagic deep ocean $CaCO_3$ sediments
(Broecker and Clark, 2009). Blooms of coccolithophores can cover up to 8 million
$km^2$ of the Earth's surface (Moore et al., 2012), and are considered to be important
drivers of biogeochemical cycling (Rost and Riebesell, 2004).
Despite intense research on coccolithophore calcification and its biogeochemical
relevance during the last decade, it is still an unresolved question why
coccolithophores calcify (Young, 1994; Raven and Crawfurd, 2012). One hypothesis
is that the layer of coccoliths surrounding the cell (coccosphere) protects the organism
from excess light and UV radiation. This notion is supported by the exceptionally
high light tolerance of the surface layer dwelling species *Emiliania huxleyi* (Nanninga
and Tyrell, 1996; Gao et al., 2009).
Physiological studies investigating the light tolerance of *E. huxleyi* showed that the
radiation wavelength matters in this context. The coccosphere does not seem to
constitute a protection against very high intensities of photosynthetically active



radiation (PAR) since non-calcifying *E. huxleyi* cells are equally resistant to
photoinhibition as their calcifying counterparts (Nanninga and Tyrrell, 1996). This is
in clear contrast to the influence of stressful ultraviolet radiation (UVR) on the cells
where results from different physiological experiments support a protective role of the
coccoliths (Gao et al., 2009; Guan and Gao, 2010; Gao et al., 2012). Protection from
UVR or high light exposures by coccoliths may either work by physically shading
intracellular organelles or by facilitating thermal dissipation through increased
non-photochemical quenching (Xu and Gao, 2011). The underlying mechanisms,
however, are not well understood and warrant further investigations.
In this study we explore in more detail how different PAR and UV radiation
(280-400 nm) treatments affect calcifying and non-calcifying *E. huxleyi* cells.
Specifically we address the question whether the coccosphere of *E. huxleyi* helps the
cells to withstand stressful levels of PAR and/or UV radiation and whether
calcification influences photochemical performance.

**2. Materials and Methods**
2.1 Materials and pre-culture conditions
Calcifying *E. huxleyi* (PML B92/11 isolated in the Raunefjord area, Bergen,
Norway) and non-calcifying cells (CCMP 2090 isolated in the South Pacific) were
used in the experiments. Both strains were grown in triplicate cultures (300 ml square
glass bottles) at 15°C in 0.2 μm filtered natural seawater (gathered from the Gulf of
Biscay) at a photon flux density of 500 μmol photons m$^{-2}$ s$^{-1}$ on a 16/8 light/dark cycle.





The natural seawater medium was enriched with 64 µmol $L^{-1}$ nitrate, 4 µmol $L^{-1}$
phosphate, f/8 concentrations of a trace metal and vitamin mixture (Guillard & Ryther
1962), and 10 nmol $kg^{-1}$ selenium. Pre-cultures and experimental incubations in
semi-continuously diluted batch cultures (>8 generations) ensured exponential growth
throughout the experiment.
2.2 Experimental setup
2.2.1 Indoor growth experiments

After pre-culture for at least 8 generations, the cells of calcifying and no-calcifying

strains were inoculated in the same glass bottles of 300 ml and cultured under the
same condition as pre-cultures, maintaining the cell concentrations at exponential
growth within a range of 3-10*$10^4$ cells/ml.
2.2.2 Outdoor growth experiments

Following the indoor growth experiment, the cells were transferred into quartz

tubes (100 ml) for the outdoor growth experiment and were exposed to natural solar
radiation at the institution's pier. The cultures were maintained outside in a
flow-through water tank, where the seawater temperature was maintained within a
range of 14-16$^{o}$C. After the cells had acclimated for 7 days under the solar radiation,
aliquots of the cell cultures were transferred to new quartz tubes filled with fresh
medium before measurements were taken. For the outdoor cultures, the cells received
60% full spectrum solar radiation (the quartz tubes wrapped with neutral density
screens). The daytime average intensities (from 7:00 am to 5:00 pm) of PAR, UV-A



and UV-B which the cells received during the outdoor experiment were about 260
µmol photons m$^{-2}$ s$^{-1}$ (about 53 W m$^{-2}$), 12.4 and 0.34 W m$^{-2}$, respectively.
2.2.3 Short-term incubation experiments
Short-term incubation experiments were carried out to test UV effects around noon
time on a cloudy day and sunny day, respectively. Three different radiation treatments
were implemented as follows: 1) Cells in uncovered quartz tubes, receiving the full
spectrum of solar radiation (above 280 nm, PAB treatment); 2) cells in quartz tubes
covered with Folex 320 (Montagefolie, Nr. 10155099, Folex, Dreieich, Germany),
exposed to UV-A and PAR (above 320 nm, PA treatment); and 3) cells receiving only
PAR (P treatment) in quartz tubes covered with Ultraphan film 395 (UV Opak,
Digefra, Munich, Germany). The transmission spectra of the quartz tubes and the
cut-off foils are given by Zheng and Gao (2009). A time-course experiment was also
conducted around noon under full solar spectrum conditions.
2.3 Absorptivity of coccoliths
We examined absorption spectra of the cells with or without coccoliths to get an
indication on how much light and/or UV are blocked by the coccosphere. Therefore,
calcified cells (Cal-C), de-calcified cells (Cal-R, see above) and cells of the naked
strain (N-Cal) were filtered onto Whatman GF/F glass fiber filters (25 mm) which
were subsequently placed at the window near the detector of a double beam
UV-VIS-NIR spectrophotometer (PerkinElmer, Lambda950, USA). The absorption of
the GF/F filter was corrected with a control filter which was soaked with particle free
culture medium (Kishino et al., 1985).



2.4 Growth measurement
Cell densities were measured during a period of 7 days with a particle counter
(Coulter Z1, Beckman). The specific growth rate was calculated as: $\mu$ ($d^{-1}$) =
($\ln N_t - \ln N_0$)/t, where $N_0$ and $N_t$ represent the cell concentrations at the beginning and
the end of the incubations and t is the incubation time in days.
2.5 Chlorophyll fluorescence measurement
Parameters of in vivo induced chlorophyll a fluorescence of photosystem II were
estimated by a phyto–pulse amplitude modulated fluorometer (Phyto-PAM, Walz).
The maximum quantum yield of PSII (Fv/Fm) was calculated as: Fv/Fm=(Fm-
Fo)/Fm; where Fo is the basal fluorescence under measuring light of 0.2 $\mu$mol
photons $m^{-2}$ $s^{-1}$ and Fm the maximal fluorescence measured with a saturating light
pulse of 5000 $\mu$mol photons $m^{-2}$ $s^{-1}$ (0.8 s) in dark-adapted (15 min) cells.
In order to compare the transmission of the same strain with or without coccoliths
and to relate this to that of the non-calcifying strain, the calcified strain was
de-calcified with HCl (1 mol/L, the final concentration is 0.01 mol/L) for 10 s and
subsequent recovery of the pH with equimolar amounts of NaOH. Photochemical
performance was measured for dark-adapted (15 min) cells in calcified, de-calcified
or non-calcifying naked cells. De-calcified cells revealed Fv/Fm values similar to
those obtained prior to de-calcification. The actinic light levels were set at 533, 1077
and 2130 $\mu$mol photons $m^{-2}$ $s^{-1}$, respectively (growth light, saturated light and
over-saturated light). Non-photochemical quenching (NPQ) was calculated as: NPQ =
($F_m - F_m'$)/$F_m'$, where $F_m$ was the maximum fluorescence yield after dark adaptation and



$F_m'$ the maximum fluorescence yield under the actinic light levels.
To determine rapid light curves (RLCs, electron transport rate vs light), the cells
were exposed to 10 different PAR levels in sequence (87, 140, 263, 382, 449, 611, 778,
993, 1195 and 1391 $\mu$mol photons m$^{-2}$ s$^{-1}$), each of which lasted for 20 s. The relative
electron transport rate (rETR) was assessed as: rETR = Yield × 0.5 × PFD, where the
yield represents the effective quantum yield of PSII ($F_v'/F_m'$); the coefficient 0.5 takes
into account that roughly 50% of all absorbed quanta reach PSII; and PFD is the
photon flux density of the actinic light ($\mu$mol m$^{-2}$ s$^{-1}$) (Genty et al., 1989).
To examine immediate photochemical responses of the cells to UV radiation, the
cells were exposed to the three different solar radiations (see above) for 60 min during
noontime under natural solar radiation. The effective quantum yield was calculated as:
$F_v'/F_m' = (F_m' - Ft) / F_m'$, where $F_m'$ and Ft are the maximal fluorescence and steady
state fluorescence in the light adapted cells, respectively.
2.6 Measurement of solar irradiances
Solar PAR was measured using a Quantum Scalar Laboratory Irradiance Sensor
(QSL-2100/ 2101, Biospherical Instruments, San Diego, USA). The measured values
were recorded every 10 s and saved on a computer. Solar UV-A and UV-B radiation
were measured with a radiometer (PMA 2100 Solar Light Co., Glenside, USA), the
mean irradiances of solar UV-A and UV-B during the experimental periods were
confirmed according to the ratios of UV-A /UV-B to PAR at the experimental location.
2.7 Statistics
The data were expressed as the means ± standard deviation (SD). Statistical



176 significance of the data was tested with software of Origin 9.0 (one way ANOVA,

177 Tukey's post-hoc test). A confidence level of 95% was used in all analyses.


179 **3 Results**

180 The coccolith layer of *E. huxleyi* absorbed both visible and UV radiation. It reduced

181 the transmission of visible radiation (400-700 nm) by 7.5%, UV-A (315-400 nm) by

182 14.1% and UVB by 18.4% (280-315 nm) relative to decalcified cells and 6.5% for

183 PAR, 6.6% for UV-A and 5.1% for UV-B, relative to non-calcifying cells (Fig. 1). The

184 specific growth rate of calcifying *E. huxleyi* strain (PML B92/11) was about 2 times

185 higher than that of the non-calcifying strain (CCMP 2090) ($P < 0.05$) when grown at

186 500 µmol photons $m^{-2}$ $s^{-1}$ of PAR under indoor conditions (Fig. 2A). Growth rates of

187 both strains were significantly ($P < 0.05$) reduced when the cells were transferred

188 outdoor and exposed to natural solar radiation. However, under outdoor conditions,

189 growth rates of calcified cells were 3.5 times higher than those of the non-calcifying

190 cells, indicating that the latter was more harmed by the solar exposure than the former

191 (Fig. 2A). The cell diameter was not significantly different in the calcified cells

192 between the indoor and outdoor conditions ($P > 0.05$), but an 18% increase was found

193 in the non-calcifying cells after they had grown under the outdoor conditions for 7

194 days ($P < 0.05$) (Fig. 2B). The maximal quantum yield (Fv/Fm) decreased when the

195 cells were transferred from indoor to the outdoor conditions, reflecting a harmful

196 effect of solar radiation. The decrease of Fv/Fm, however, was much more

197 pronounced in the non-calcifying cells (27%) compared to calcifying cells (11%) (Fig.



2C).
Calcified cells had significantly higher rETR, higher apparent light use efficiency
(α), and higher maximal electron transport rate (rETR$_{max}$), but significantly lower
light saturation parameters (Ik). The de-calcified cells of the calcifying strain showed
a remarkable decrease of rETRmax ($P < 0.05$), but did not show obvious changes in α
and Ik (Fig. 3, Table 1). Increased actinic light levels (acclimating light during the
fluorescence measurement) led to higher NPQ in both the calcifying and
non-calcifying strain (Fig. 4). Furthermore, calcified cells showed higher NPQ values
compared to non-calcifying cells ($p < 0.05$).
When exposed to full spectrum solar radiation, the quantum yield of calcified cells
showed no significant change during the first 30 min ($P > 0.05$). After 30 minutes,
quantum yield quickly dropped from about 0.35 to 0.22 for ~20 min ($P < 0.05$)
followed by a slight recovery in the last 25 minutes. A similar trend was observed in
the de-calcified cells with the key difference that the sharp decrease already happened
during the first 10 min. Quantum yield of the non-calcifying cells decreased
constantly for the first 50 minutes and remained at the low level thereafter (Fig. 5).
No effect of the radiation treatment (P, PA and PAB radiation) on the quantum yield
of calcified cells was observed after the cells grown under indoor condition were
transferred to outdoor solar radiation for 1h exposure (very cloudy day, average PAR,
UV-A and UV-B were 481μmol photons m$^{-2}$ s$^{-1}$, 22.1 and 0.7 W m$^{-2}$, respectively) ($P >$
0.05). Quantum yield was significantly higher in the non-calcifying cells, however,
when they were exposed to UVA radiation (PA vs. P treatment, $P < 0.05$ Fig. 6A).





Similar responses were observed when the same test was done on a sunny day with
average PAR, UV-A and UV-B of 1605 µmol photons m$^{-2}$ s$^{-1}$, 69 and 2.4 W m$^{-2}$,
respectively. Here, the quantum yield of the calcified cells showed no significant
difference between the different light treatments but it decreased significantly under
PAB treatment compared to P treatments in the non-calcifying cells ($P < 0.05$) (Fig.
6B).

**227    4 Discussion**

Various hypotheses were proposed for the possible functions of coccoliths, but none
of them is supported by sufficient evidence (Young, 1994; Raven and Crawfurd,
2012). One important function of coccoliths for surface-dwelling species such as *E.*
*huxleyi* could be the protection against high photon flux densities, especially UV
radiation (Berge, 1962; Young, 1994; Gao et al., 2009).
Some of our results support this hypothesis. The growth rate of the calcified cells of
*E. huxleyi* grown under indoor conditions was about 2 times higher than that of naked
cells. This difference came out even stronger, with growth rates 3.5 times higher in
calcified versus naked cells, when the cells were exposed to full spectrum solar
radiation (Fig. 2A). This could potentially be attributed to the screening of PAR,
UV-A, and UV-B by coccoliths. Although the daytime PAR of solar radiation was
reduced to about half of the light level of the indoor test, noon time PAR levels were
higher than 500µmol photons m$^{-2}$ s$^{-1}$, and the presence of UV could lead to more
harms to the naked cells. Light protection by coccoliths is further supported by the



Fv/Fm measurements. The maximum photochemical efficiency of PSII was only
slightly reduced in calcified cells but significantly decreased in non-calcifying cells
when they were exposed to natural solar PAR and UV radiation (Fig. 2C).
Furthermore, photochemical performance of de-calcified cells decreased significantly
faster and stronger with time compared calcified cells (Fig. 5).

The diameter of calcified cells did not significantly change when they were

exposed to the full spectrum of solar radiation. The diameter of the non-calcifying
cells, however, increased significantly (Fig. 2B). Perhaps, the non-calcifying cells
experienced more DNA damage and so did not enter the S phase regularly (Buma et
al., 2000). Alternatively, it may reflect a strategy to acclimatize to stressful solar UV
radiation since it is well known that smaller cells are usually more sensitive to UV
than their larger counterparts (Garcia-Pichel, 1994; Laurion and Vincent, 1998). Some
field and laboratory studies showed increased cell size with increased UV exposures
(Buma et al., 2000), which can be interpreted as adaptive or acclimation mechanism
for protecting the cells against UV radiation.   .

Several studies found that coccoliths do not protect *E. huxleyi* from excess PAR

(Nanninga and Tyrrell, 1996; Houdan et al., 2005; Trimborn et al., 2007). However,
UV radiation was not considered in these experiments. Our results showed that the
non-calcifying cells were more sensitive to full spectrum solar radiation than calcified
cells and even in the same strain, the photochemical performance of de-calcified cells
decreased significantly when comparing the calcified cells. This suggests that
coccoliths efficiently protect the cells from solar UV radiation.



On the other hand, *E. huxleyi* appears to be more sensitive to UV-B irradiances than
other phytoplankton species, and its growth rate and physiological performances were
highly inhibited by UV radiation (Peletier et al., 1996; Buma et al., 2000; Xu et al.,
2011). However, competition tests for community changes are rare, and longer-term
experiments with less extreme UVR would be more ecologically and evolutionarily
relevant (Raven and Crawfurd, 2012). In our work, UVR had no significant effect on
the quantum yield of calcified cells regardless of high or low light condition but it
showed inhibition in non-calcifying cells when they were exposed to high solar light
(Fig. 6A, B). This provides further evidence for protection by coccoliths against UV
radiation.
On the cloudy day, no significant difference was observed among the treatments for
the calcifying cells; on the sunny day, under the fluctuating light (data not shown)
calcifying cells manage to refurbish damage to their photosynthetic apparatus by
balancing damage and repair (Gao et al., 2007). For the non-calcifying cells, on the
other hand, UV damage was not effectively repaired, leading to the observed negative
effect on photosynthetic performance.
In conclusion, the coccoliths of calcifying *E. huxleyi* play an important role in
protecting this species against harmful solar radiation especially UV-A and UV-B .
The reported absence of photoinhibition in this alga at high light levels is most likely
connected to the photoprotective role played by the coccosphere of *E. huxleyi*. With
shoaling of the upper mixed layer (UML) caused by global warming and progressive
ocean acidification, reduced thickness or the number of coccoliths (Gao et al., 2009;



De Bodt et al., 2010), cells of *E. huxleyi* living within the UML would be impacted
due to increased daily exposures to solar radiation.

**Acknowledgements**
This study was supported by National Natural Science Foundation (41430967;
41476097; 41120164007), State Oceanic Administration (National Programme on
Global Change and Air-Sea Interaction, GASI-03-01-02-04), Joint project of National
Natural Science Foundation of China and Shandong province (No. U1406403),
Strategic Priority Research Program of Chinese Academy of Sciences (No.
XDA1102030204). Visit of KG to Kiel was supported by DAAD.

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

**Figure captions**
**Figure 1.** Transmission spectra of cells with (Cal-C, calcifying strain) and without
(Cal-R, calcifying strain with coccoliths removed artificially) coccolith cover and
non-calcifying (N-Cal) cells of *Emiliania huxleyi*.

**Figure 2.** The specific growth rate ($\mu$) (A), diameter (B) and maximum quantum yield



(C) of PSII (Fv/Fm) of the calcified (Cal-C) and non-calcifying (N-Cal) cells of *E.*
*huxleyi* grown in indoor and outdoor conditions. Different letters represent significant
difference between the indoor and outdoor experiments. Different horizontal lines
represent significant difference between the different strains.

**Figure 3.** The relative electron rate (rETR) of coccolith-covered (Cal-C),
coccolith-removed (Cal-R) and non-calcifying (N-Cal) cells of *E. huxleyi* grown
under indoor conditions as function of PAR. The cells had been grown for 12-22
generations under 500 $\mu$mol photons m$^{-2}$ s$^{-1}$ of PAR.

**Figure 4.** The non-photochemical quenching (NPQ) of coccolith-covered (Cal-C) and
non-calcifying (N-Cal) cells of *E. huxleyi* grown under indoor conditions. Different
letters represent significant difference among the light levels. Different horizontal
lines represent significant difference among the different type cells.

**Figure 5.** The time course of quantum yield of coccolith-covered (Cal-C),
coccolith-removed (Cal-R) and non-calcifying (N-Cal) cells of *E. huxleyi* under full
spectrum solar radiation (noontime, average PAR, UV-A and UV-B were 1082$\mu$mol
photons m$^{-2}$ s$^{-1}$, 48.1 and 1.6 W m$^{-2}$, respectively).

**Figure 6.** The change of quantum yield of the calcified (Cal-C) and non-calcifying
(N-Cal) cells of *E. huxleyi* when transferred from indoor to outdoor conditions, being
exposed to PAR alone (P), PAR+UVA(PA) and PAR+UVA+B(PAB) for 60 min at
around noon time. A, measured under a cloudy day (average PAR, UV-A and UV-B
were 481$\mu$mol photons m$^{-2}$ s$^{-1}$, 22.1 and 0.7 W m$^{-2}$, respectively); B, measured under





a sunny day (average PAR, UV-A and UV-B were 1605 µmol photons $m^{-2}$ $s^{-1}$, 69 and
2.4 $W\,m^{-2}$).   Different letters represent significant difference among the light
treatments. Different horizontal lines represent significant difference between the
different strains.






















Table 1. Photosynthetic parameters of relative electron transport rate (Figure 3) as a
function of PAR, different letters represent significant difference (P<0.05) among the
treatments.

| | $\alpha$ | $rETR_{max}$ | $I_k$ |
|---|---|---|---|
| Cal-C | 0.23±0.02[a] | 90.6±9.0[a] | 1010.8±95.0[a] |
| Cal-R | 0.20±0.01[a] | 73.5±3.5[b] | 986.3±27.4[a] |
| N-Cal | 0.17±0.02[b] | 42.3±8.5[c] | 621.8±111.1[b] |






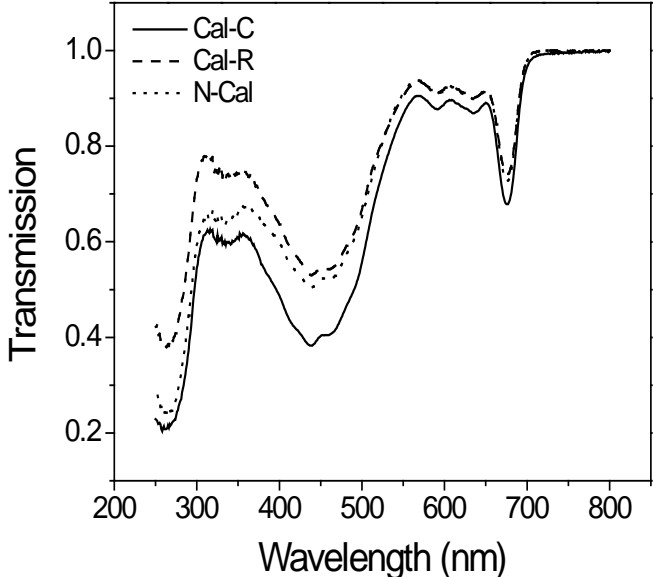




Fig. 1




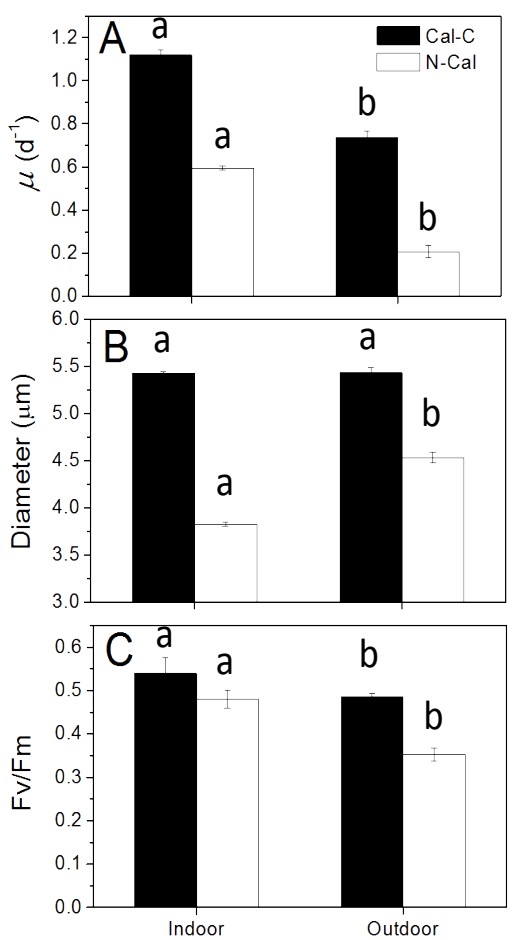



Fig. 2







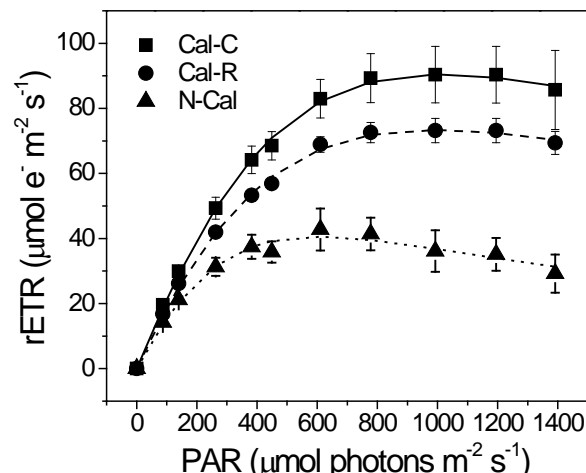




Fig. 3


















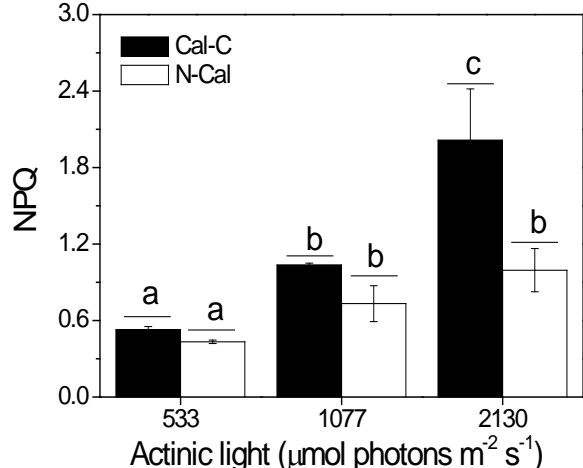





Fig. 4









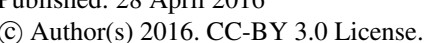



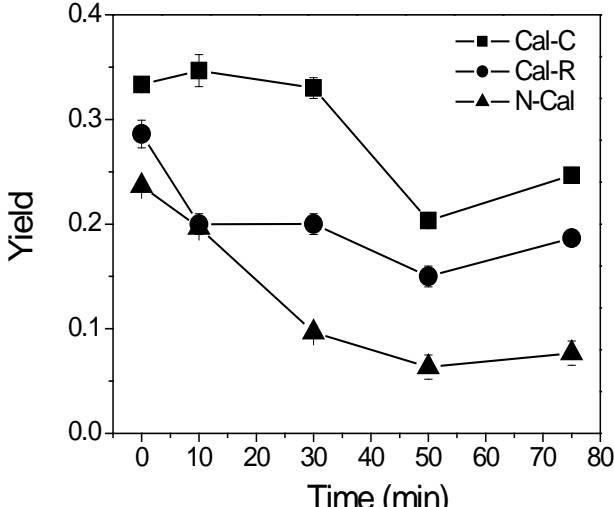




Fig. 5










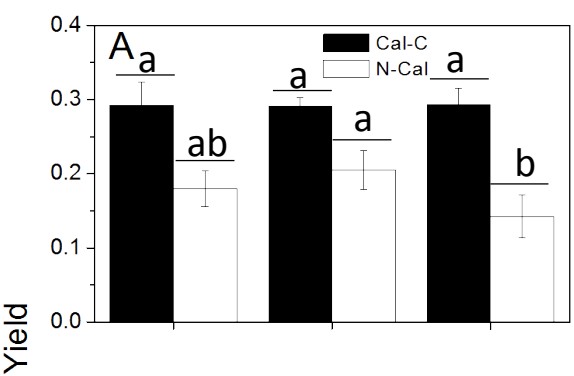



Fig. 6






