# Peer review of "The role of coccoliths in protecting *Emiliania huxleyi* against stressful light and UV radiation"

_Biogeosciences, 2016_

## Referee Comment (RC1) · D. Campbell (Referee) · 10 May 2016

The authors tested the growth and photophysiological responses of Emiliania huxleyi to PAR and UV in the presence of coccoliths, after removal of coccoliths and in a strain that lacks coccoliths.

The data show that presence of coccoliths renders the cells less susceptible to inhibition by UV, and increases their capacity for non-photochemical quenching.

The manuscript presents a tidy study on an important question, and is appropriate for BioGeoScience.

I offer a few minor wording and reference comments for the author's consideration. best regards, Doug Campbell

Abstract: Fine.

"...since decades..." is not incorrect, but is idiomatically odd. I suggest "...for decades...".

Introduction: "This notion is supported by the exceptionally high light tolerance of the surface layer dwelling species Emiliania huxleyi (Nanninga and Tyrell 1996; Gao et al., 2009)"

Geider's group had a paper: Ragni M, Airs RL, Leonardos N, Geider RJ. 2008. PHOTOINHIBITION OF PSII IN EMILIANIA HUXLEYI (HAPTOPHYTA) UNDER HIGH LIGHT STRESS: THE ROLES OF PHOTOACCLIMATION, PHOTOPROTECTION, AND PHOTOREPAIR. Journal of Phycology 44: 670–683.

and we had a paper: Loebl M, Cockshutt AM, Campbell DA, Finkel ZV. 2010. Physiological basis for high resistance to photoinhibition under nitrogen depletion in Emiliania huxleyi. Limnology and Oceanography 55: 2150–2160.

both showing that the high PAR tolerance of E. hux related to very strong repair capacities, rather than intrinsic resistance to photoinactivation, per se. It would be worth noting that UV is a strong inhibitor of PSII repair, as well as acting through direct inhibition of PSII. So it could be that the coccoliths protect PSII repair from UV inhibition.

I now read you briefly make this point in the discussion, citing Gao 2007.

Materials & Methods: Fine

Results: "Photochemical performance was measured for dark-adapted (15 min) cells in calcified, de-calcified or non-calcifying naked cells"

The table and figure abbreviation Cal-R does not obviously suggest 'de-calcified'. Why not 'D-Cal' or 'Cal-D'?. More generally, why erect abbreviations? Why not just write out 'Calcified', 'De-calcified', and 'Naked'?

In the text the naked strain is sometimes called naked, or sometimes 'non-calcifying'.

Unify the terminology; pick a single name for each cell condition and use it throughout.

---

## Referee Comment (RC2) · Anonymous Referee #2 · 2 Jun 2016

Coccolithophores are an ecologically important group of marine phytoplankton that characteristically produce calcium carbonate plates (liths) internally and then secrete them to the cell surface. Exactly why coccolithophores produce liths has been the subject of considerable debate, with a range of possibilities raised. The manuscript by Xu et al has set out to test the hypothesis that the calcite from which the liths are constructed will absorb enough UV radiation to protect the cells from damage.

The approach the authors have used is to compare the UV sensitivity (measured as growth, quantum yield of PSII and relative electron transport characteristics including non-photochemical quenching) in a calcified strain, a non-calcifying strain and the calcifying strain with the liths removed (which could be more clearly termed 'decalcified' in the text and figures/tables). The experimental approach is sound and the results are presented clearly and discussed thoroughly.

I really only have some minor points the authors might like to consider.

P 2 line 26: 'for' not 'since'

P 3 line 52: delete 'by'

P 4 lines 72/72: shading effects could include scattering of light which is certainly a feature of coccolithophore blooms

P 4 lines 73/74: a bit pedantic I know, but strictly speaking it is hard to see how liths could stimulate NPQ – presumably liths are affecting the light climate in some way that leads to up-regulation of NPQ

P 9 line 180 et seq.: I assume some of this loss of transmittance could be due to scattering by liths rather than absorbance? In Fig 1, the non calcifying strain has a lower transmittance in the UV than does the calcifying strain without liths - could this be because the non-calcifying strain employs another strategy (such as inducing UV-screening compounds) to ameliorate UVB?

P 10 lines 201-203: A significant decrease in rETRmax would, if alpha is unaffected, be expected to show a decrease in Ik (given the relationship Ik=rETRmax/alpha) so the lack of an effect on Ik seems odd. In fact there does seem to be a difference but it is statistically non-significant.

P 11 discussion lines 233- 234: While I don't disagree with the authors conclusions, you have to be a bit careful in making claims that the liths are causing the differences in growth rate between the naked and calcifying strain based on only one strain of each phenotype (certainly when grown indoors). Would all naked strains grow more slowly than all calcified strains? Remember also that calcification leads to internal generation of $CO_2$ - although it has been shown elegantly by Bach et al 2013 that there is no obligatory coupling between calcification and photosynthesis. The differential impact of UVB on growth is though compelling!

---

## Short Comment (SC1) · 24 Jun 2016

Referee #1 The authors tested the growth and photophysiological responses of Emiliania huxleyi to PAR and UV in the presence of coccoliths, after removal of coccoliths and in a strain that lacks coccoliths.The data show that presence of coccoliths renders the cells less susceptible to inhibition by UV, and increases their capacity for non-photochemical quenching. The manuscript presents a tidy study on an important question, and is appropriate for BioGeoScience.I offer a few minor wording and reference comments for the author's consideration. best regards, Doug Campbell

Abstract: Fine. "...since decades..." is not incorrect, but is idiomatically odd. I suggest "...for decades...". Response: corrected as suggested. Introduction: "This notion is supported by the exceptionally high light tolerance of the surface layer dwelling species

Emiliania huxleyi (Nanninga and Tyrell 1996; Gao et al., 2009)"Geider's group had a paper: Ragni M, Airs RL, Leonardos N, Geider RJ. 2008. PHOTOINHIBITION OF PSII IN EMILIANIA HUXLEYI (HAPTOPHYTA) UNDER HIGH LIGHT STRESS: THE ROLES OF PHOTOACCLIMATION,PHOTOPROTECTION, AND PHOTOREPAIR. Journal of Phycology 44: 670–683. and we had a paper: Loebl M, Cockshutt AM, Campbell DA, Finkel ZV. 2010. Physiological basis for high resistance to photoinhibition under nitrogen depletion in Emiliania huxleyi. Limnology and Oceanography 55: 2150–2160. both showing that the high PAR tolerance of E. hux related to very strong repair capacities, rather than intrinsic resistance to photoinactivation, per se. It would be worth noting that UV is a strong inhibitor of PSII repair, as well as acting through direct inhibition of PSII. So it could be that the coccoliths protect PSII repair from UV inhibition. I now read you briefly make this point in the discussion, citing Gao 2007. Response: We thank the reviewer for this constructive comments. We added a sentence in the last paragraph of introduction and cited extra references. The mentioned references above were cited in the introduction and discussion at lines 63 and 279. Materials & Methods: Fine Results: "Photochemical performance was measured for dark-adapted (15 min) cells in calcified, de-calcified or non-calcifying naked cells" The table and figure abbreviation Cal-R does not obviously suggest 'de-calcified'. Why not 'D-Cal' or 'Cal-D'?. More generally, why erect abbreviations? Why not just write out 'Calcified', 'De-calcified', and 'Naked'? In the text the naked strain is sometimes called naked, or sometimes 'non-calcifying'. Unify the terminology; pick a single name for each cell condition and use it throughout.

Response: As you suggested, we corrected them as 'Calcified', 'De-calcified', and 'Naked'

Anonymous Referee #2 Coccolithophores are an ecologically important group of marine phytoplankton that characteristically produce calcium carbonate plates (liths) internally and then secrete them to the cell surface. Exactly why coccolithophores produce liths has been the subject of considerable debate,

with a range of possibilities raised. The manuscript by Xu et al has set out to test the hypothesis that the calcite from which the liths are constructed will absorb enough UV radiation to protect the cells from damage. The approach the authors have used is to compare the UV sensitivity (measured as growth, quantum yield of PSII and relative electron transport characteristics including non-photochemical quenching) in a calcified strain, a non-calcifying strain and the calcifying strain with the liths removed (which could be more clearly termed 'decalcified' in the text and figures/tables). Response: Corrected as "de-calcified" The experimental approach is sound and the results are presented clearly and discussed thoroughly. I really only have some minor points the authors might like to consider. P 2 line 26: 'for' not 'since' Response: Corrected. P 3 line 52: delete 'by' Response: deleted.

P 4 lines 72/72: shading effects could include scattering of light which is certainly a feature of coccolithophore blooms Response: We added this in the text and cited two papers to support it at lines 71-74. P 4 lines 73/74: a bit pedantic I know, but strictly speaking it is hard to see how liths could stimulate NPQ – presumably liths are affecting the light climate in some way that leads to up-regulation of NPQ Response: We think it is possible that NPQ is indirectly affected, which coincides with calcification. This sentence was deleted in the text. P 9 line 180 et seq.: I assume some of this loss of transmittance could be due to scattering by liths rather than absorbance? In Fig 1, the non calcifying strain has a lower transmittance in the UV than does the calcifying strain without liths - could this be because the non-calcifying strain employs another strategy (such as inducing UVscreening compounds) to ameliorate UVB? Response: We used a double beam UV-VIS-NIR spectrophotometer (PerkinElmer, Lambda950, USA) to obtain the absolute absorbance of liths, so that all the scattered light was recaptured. It might be one of strategies against UV damage for non calcifying strain to synthesize UV screening compounds, and we added this in the discussion at lines 256-259.

P 10 lines 201-203: A significant decrease in rETRmax would, if alpha is unaffected, be expected to show a decrease in Ik (given the relationship Ik=rETRmax/alpha) so the

lack of an effect on Ik seems odd. In fact there does seem to be a difference but it is statistically non-significant. Response: Considering the relationship Ik=rETRmax/alpha, it would be true. In our results, there are differences in Ik between Calcified and Decalcified cells, which is only statistically different at P>0.05. Since alpha is lower in Decalcified cells than that in calcified cells, which is not statistically different at P<0.05, Ik=rETRmax/alpha could not statistically be different.

P 11 discussion lines 233- 234: While I don't disagree with the authors conclusions, you have to be a bit careful in making claims that the liths are causing the differences in growth rate between the naked and calcifying strain based on only one strain of each phenotype (certainly when grown indoors). Would all naked strains grow more slowly than all calcified strains? Remember also that calcification leads to internal generation of CO2 - although it has been shown elegantly by Bach et al 2013 that there is no obligatory coupling between calcification and photosynthesis. The differential impact of UVB on growth is though compelling! Response: We have revised the conclusion, now it reads "In conclusion, the coccoliths of calcified E. huxleyi play an important role in protecting this species against harmful solar radiation especially UV-A and UV-B. Since our data were only based on only one strain of each phenotype, cautions should be taken when drawing conclusion about the differential responses of calcifying and naked strains. In addition, since calcification leads to internal generation of CO2, such an increased availability of CO2 might aid to enhance growth, though it has been shown that there is no obligatory coupling between calcification and photosynthesis (Bach et al., 2013). Nevertheless, the reported absence of photoinhibition in this alga at high light levels appears to be connected to the coccosphere of E. huxleyi or its calcification process. In view of the ecological implications, shoaling of the upper mixed layer (UML) caused by global warming and progressive ocean acidification that reduces thickness or the number of coccoliths (Gao et al., 2009; De Bodt et al., 2010) could threaten cells of E. huxleyi living within the UML due to increased daily exposures to solar radiation. "

---

## Author Response (AR1)

**Responses to the comments of reviewers**

**Referee #1**

The authors tested the growth and photophysiological responses of Emiliania huxleyi to PAR and UV in the presence of coccoliths, after removal of coccoliths and in a strain that lacks coccoliths.The data show that presence of coccoliths renders the cells less susceptible to inhibition by UV, and increases their capacity for non-photochemical quenching. The manuscript presents a tidy study on an important question, and is appropriate for BioGeoScience.I offer a few minor wording and reference comments for the author's consideration.

best regards, Doug Campbell

Abstract: Fine.

"...since decades..." is not incorrect, but is idiomatically odd. I suggest "...for decades...".

Response: corrected as suggested.

Introduction: "This notion is supported by the exceptionally high light tolerance of the surface layer dwelling species Emiliania huxleyi (Nanninga and Tyrell 1996; Gao et al., 2009)"Geider's group had a paper: Ragni M, Airs RL, Leonardos N, Geider RJ. 2008. PHOTOINHIBITION OF PSII IN EMILIANIA HUXLEYI (HAPTOPHYTA) UNDER HIGH LIGHT STRESS: THE ROLES OF PHOTOACCLIMATION,PHOTOPROTECTION, AND PHOTOREPAIR. Journal of Phycology 44: 670–683. and we had a paper: Loebl M, Cockshutt AM, Campbell DA, Finkel ZV. 2010. Physiological basis for high resistance to photoinhibition under nitrogen depletion in Emiliania huxleyi. Limnology and Oceanography 55: 2150–2160. both showing that the high PAR tolerance of E. hux related to very strong repair capacities, rather than intrinsic resistance to photoinactivation, per se. It would be worth noting that UV is a strong inhibitor of PSII repair, as well as acting through direct inhibition of PSII. So it could be that the coccoliths protect PSII repair from UV inhibition. I now read you briefly make this point in the discussion, citing Gao 2007.

Response: We thank the reviewer for this constructive comments.   We added a sentence in the last paragraph of introduction and cited extra references. The mentioned references above were cited in the introduction and discussion at lines 75 and 355.

Materials & Methods: Fine

Results: "Photochemical performance was measured for dark-adapted (15 min) cells in calcified, de-calcified or non-calcifying naked cells"

The table and figure abbreviation Cal-R does not obviously suggest 'de-calcified'. Why not 'D-Cal' or 'Cal-D'?. More generally, why erect abbreviations? Why not just write out 'Calcified', 'De-calcified', and 'Naked'? In the text the naked strain is sometimes called naked, or sometimes 'non-calcifying'. Unify the terminology; pick a single name for each cell condition and use it throughout.

Response: As you suggested, we corrected them as 'Calcified', 'De-calcified', and 'Naked'

**Anonymous Referee #2**

Coccolithophores are an ecologically important group of marine phytoplankton that characteristically produce calcium carbonate plates (liths) internally and then secrete them to the cell surface. Exactly why coccolithophores produce liths has been the subject of considerable debate, with a range of possibilities raised. The manuscript by Xu et al has set out to test the hypothesis that the calcite from which the liths are constructed will absorb enough UV radiation to protect the cells from damage.

The approach the authors have used is to compare the UV sensitivity (measured as growth, quantum yield of PSII and relative electron transport characteristics including non-photochemical quenching) in a calcified strain, a non-calcifying strain and the calcifying strain with the liths removed (which could be more clearly termed 'decalcified' in the text and figures/tables).

Response: Corrected as "de-calcified"

The experimental approach is sound and the results are presented clearly and discussed thoroughly. I really only have some minor points the authors might like to consider.

P 2 line 26: 'for' not 'since'

Response: Corrected.

P 3 line 52: delete 'by'

Response: deleted.

P 4 lines 72/72: shading effects could include scattering of light which is certainly a feature of coccolithophore blooms

Response: We added this in the text and cited two papers to support it at lines 84-87.

P 4 lines 73/74: a bit pedantic I know, but strictly speaking it is hard to see how liths could stimulate NPQ – presumably liths are affecting the light climate in some way that leads to up-regulation of NPQ

Response: We think it is possible that NPQ is indirectly affected, which coincides with calcification. This sentence was deleted in the text.

P 9 line 180 et seq.: I assume some of this loss of transmittance could be due to scattering by liths rather than absorbance? In Fig 1, the non calcifying strain has a lower transmittance in the UV than does the calcifying strain without liths - could this be because the non-calcifying strain employs another strategy (such as inducing UVscreening compounds) to ameliorate UVB?

Response: We used a double beam UV-VIS-NIR spectrophotometer (PerkinElmer, Lambda950, USA) to obtain the absolute absorbance of liths, so that all the scattered light was recaptured. This was added in the Methods (lines 166-167). It might be one of strategies against UV damage for non calcifying strain to synthesize UV screening compounds, and we added this in the discussion at lines 320-323.

P 10 lines 201-203: A significant decrease in rETRmax would, if alpha is unaffected, be expected to show a decrease in Ik (given the relationship Ik=rETRmax/alpha) so the lack of an effect on Ik seems odd. In fact there does seem to be a difference but it is statistically non-significant.

Response: Considering the relationship Ik=rETRmax/alpha, it would be true. In our results, there are differences in Ik between Calcified and De-calcified cells, which is only statistically different at P>0.05. Since alpha is lower in De-calcified cells than that in calcified cells, which is not statistically different at P<0.05, Ik=rETRmax/alpha could not statistically be different.

P 11 discussion lines 233- 234: While I don't disagree with the authors conclusions, you have to be a bit careful in making claims that the liths are causing the differences in growth rate between the naked and calcifying strain based on only one strain of each phenotype (certainly when grown indoors). Would all naked strains grow more slowly than all calcified strains? Remember also that calcification leads to internal generation of CO2 - although it has been shown elegantly by Bach et al 2013 that there is no obligatory coupling between calcification and photosynthesis. The differential impact of UVB on growth is though compelling!

[revised manuscript text omitted]

**Figure 6.** The change of quantum yield of the calcified and naked cells of *E. huxleyi*

when transferred from indoor to outdoor conditions, being exposed to PAR alone (P),

PAR+UVA(PA) and PAR+UVA+B(PAB) for 60 min at around noon time. A, measured under a cloudy day (average PAR, UV-A and UV-B were 481µmol photons

$m^{-2} s^{-1}$, 22.1 and 0.7 W $m^{-2}$, respectively); B, measured under a sunny day (average

PAR, UV-A and UV-B were 1605 µmol photons $m^{-2} s^{-1}$, 69 and 2.4 W $m^{-2}$).  Different letters represent significant difference among the light treatments. Different horizontal lines represent significant difference between the different strains.

Table 1. Photosynthetic parameters of relative electron transport rate (Figure 3) as a function of PAR, different letters represent significant difference (P<0.05) among the calcified, de-calcified and naked cells.

| | $\alpha$ | $rETR_{max}$ | $I_k$ |
|---|---|---|---|
| Calcified | $0.23\pm0.02^a$ | $90.6\pm9.0^a$ | $1010.8\pm95.0^a$ |
| De-calcified | $0.20\pm0.01^a$ | $73.5\pm3.5^b$ | $986.3\pm27.4^a$ |
| naked | $0.17\pm0.02^b$ | $42.3\pm8.5^c$ | $621.8\pm111.1^b$ |

Fig. 1

[Figure]

                              Fig. 2

Fig. 3

[Figure]

                                Fig. 4

[Figure]

                      Fig. 5

[Figure]

                                  Fig. 6